# Hawkes process revisited: balancing interpretability and flexibility with contextualized event embeddings and a neural impact kernel

## Abstract

The Hawkes process (HP) is commonly used to model event sequences with self-reinforcing dynamics, including electronic health records, stock trades, and social media interactions. Traditional HPs capture self-reinforcement via parametric impact functions that can be inspected to understand how each event modulates the intensity of others. Neural network-based HPs offer greater flexibility, resulting in improved fit and prediction performance, but at the cost of interpretability, which can be critical in medicine and other high-stakes settings. In this work, we aim to understand and improve upon this tradeoff. We propose a novel HP formulation in which impact functions are modeled by defining a flexible impact kernel, instantiated as a neural network, in event embedding space, which allows us to model large-scale event sequences with many event types. This approach is more flexible than traditional HPs, because we do not assume a particular parametric form for the impact functions, yet more interpretable than other neural network approaches, because self-reinforcing dynamics are still entirely captured by the impact kernel, which can be inspected. If needed, our approach allows us to trade interpretability for flexibility by contextualizing the event embeddings with transformer encoder layers. Results show that our method accurately recovers impact functions in simulations and achieves competitive performance on real-world datasets even without transformer layers. This suggests that our flexible impact kernel is often sufficient to capture self-reinforcing dynamics effectively, implying that interpretability can be maintained without loss of performance.

## 1 Introduction

The Hawkes process (HP) is a powerful tool for modeling event sequences with self-reinforcing dynamics, making it applicable to domains such as electronic health records (EHRs) (Wang et al., 2018), stock trading (Bacry et al., 2015), and social media interactions (Yang & Leskovec, 2011). Unlike the Poisson process, which assumes events occur independently over time with a constant intensity, the HP captures the influence of past events on the likelihood of future occurrences by introducing self-excitation. This self-exciting property allows the HP to model the clustering behavior frequently observed in real-world event data (Hawkes, 1971; Ogata, 1988). For instance, in EHRs, a patient's sequence of diagnoses, prescriptions, and lab tests can reflect how each medical event influences subsequent treatments and health outcomes. While much research focuses on predicting future events (Shchur et al., 2019; Mei & Eisner, 2017), we aim to build interpretable models that reveal how events influence one another, shedding light on Granger causality.

However, capturing the self-reinforcement inherent in traditional Hawkes Processes (HPs) often relies on parametric impact functions, which may fall short in representing the intricate patterns found in real-world data (Linderman & Adams, 2014). To overcome these limitations, researchers have increasingly turned to neural networks integrated with point processes, fostering the development of more adaptable models. For instance, the Neural Hawkes Process leverages recurrent neural networks to model conditional intensity functions, effectively capturing non-linear and long-range temporal dependencies (Mei & Eisner, 2017). Similarly, the Transformer Hawkes Process utilizes self-attention mechanisms to adeptly manage event dependencies within sequences of varying lengths (Zuo et al., 2020). Additionally, the Intensity-Free method for learning Temporal Point

Processes (TPP) offers a novel approach by directly learning the conditional density of inter-event times, thereby circumventing the computationally intensive integrals required by traditional intensity functions (Shchur et al., 2019). Another noteworthy advancement is the Neural Spatio-Temporal Point Processes (ODETPP), which models the evolution of hidden states through neural ordinary differential equations, simplifying previous models by focusing exclusively on temporal dependencies without incorporating spatial components (Chen et al., 2020). While these innovations provide richer representations of intensity functions, they often do so at the expense of interpretability (Xu et al., 2020).

Many of these methods, while powerful, relax so many of the Hawkes process's original assumptions that they lose key features and become black-box models (Linderman & Adams, 2014). This lack of transparency makes it difficult to interpret event interactions or uncover Granger causality, which can be critical in fields like medicine, marketing, and biology (Eichler et al., 2017). In these domains, understanding the relationships between events is as important as predicting the events themselves. For example, in healthcare, identifying how medical events relate can guide clinical research and improve patient care (Liu et al., 2019), and emerging regulations governing use of models for clinical decision-making emphasize interpretability(U.S. Food and Drug Administration, 2023). In business, uncovering how marketing events drive sales informs strategy (Hawkes, 2018). In animal behavior studies, finding associations between behaviors can reveal biological mechanisms (Ward et al., 2022). To serve these needs, we require models that balance flexibility with transparency.

In this work, we aim to understand and improve the tradeoff between interpretability and flexibility in HP models. To do this, we propose a novel HP formulation in which impact functions are modeled by defining a flexible, neural network-based impact kernel in event embedding space. Our formulation maintains the core formulation of the Hawkes process, combining a baseline intensity with an impact function summed over all previous events, and thereby retains the Hawkes process's key properties, such as positive intensity and additive influence. However, to improve flexibility, we replace the traditional exponential decay assumption(Hawkes, 1971) with neural network-driven impact functions, allowing for more nuanced modeling of event dependencies without sacrificing interpretability.

By working in embedding space, we limit the dimensionality of our impact kernel, which allows us to model large-scale event sequences with tens of thousands of event types. Each dimension in this space represents a broader event topic, with the impact kernel capturing the relationships between these topics. This makes large-scale event modeling computationally feasible while enhancing interpretability, as the interactions are understood at the topic level.

In some cases, it may be desirable to trade interpretability for greater flexibility. Our approach allows users to explore this tradeoff directly by adding optional transformer encoder layers to contextualize the embeddings of each event in a given sequence based on the previous history of events. Adding these layers reduces interpretability and sacrifices the property of additive influence, but in principle, it also allows the model to capture more complex dependencies between event types. Importantly, however, our results show that this sacrifice is rarely necessary, because our flexible impact kernel alone is sufficient to capture the dynamics of most real-world sequences.

In summary, our contributions are:

1. **Novel Hawkes Process Formulation:** We introduce a generalized Hawkes process where impact functions are defined via a flexible, neural network-based impact kernel within an event embedding space.

2. **Controlled Tradeoff Between Interpretability and Flexibility:** We develop a method to explicitly manage the balance between interpretability and model complexity by incorporating transformer encoder layers to contextualize event embeddings based on the historical sequence of events.

3. **Maintaining Interpretability Without Sacrificing Performance:** We demonstrate that in real-world settings, transformer encoder layers are often unnecessary to achieve state-of-the-art performance, thereby maintaining interpretability without compromising the model's effectiveness.

## 2 RELATED WORK

**Generalized Hawkes Process** Traditional Hawkes processes are limited in their ability to model complex event dynamics due to their reliance on simple exponential decay for event intensity. To address these limitations, neural network-based extensions have been developed. One key advancement is the Neural Hawkes Process (NHP), which generates events sequentially using recurrent neural networks. NHP replaces the traditional intensity function with one parameterized by a recurrent model, allowing it to better capture dependencies between events.

Another generalized Hawkes process is the attention-based Hawkes process. One of the classic example is the Transformer Hawkes Process(THP)(Zuo et al., 2020). THP leverages attention mechanisms to model event intensity, using transformer-based encodings to incorporate both temporal and contextual information. It enhances the intensity function by introducing terms to account for event timings and baseline effects. By doing so, THP captures the influence of past events on future ones in a more flexible and context-aware manner. Like other generalized Hawkes processes, both NHP and THP are trained to maximize the likelihood of observed event sequences, enabling them to model more complex event interactions.

**Self-Attentive Hawkes Process (SAHP)** The Self-Attentive Hawkes Process (SAHP) (Zhang et al., 2020) extends the traditional Hawkes process by incorporating a self-attention mechanism. This enhancement allows for better modeling of complex event dependencies and improves interpretability. SAHP quantifies the influence of historical events on future ones using attention weights, providing an interpretable measure of how past events affect subsequent occurrences. By accumulating these attention weights, SAHP effectively quantifies the statistical influence between different event types, making it valuable for both predicting event sequences and explaining event relationships.

However, SAHP represents a significant departure from the classical Hawkes process, which models the impact of events through explicit time-decaying influence functions. In contrast, SAHP relies on learned attention weights without explicitly capturing the temporal dynamics of influence decay inherent in the original impact function.

## 3 METHODS

### 3.1 THE HAWKES PROCESS

Let us denote an event sequence as $\mathcal{S} = \{(t_1, k_1), (t_2, k_2), \ldots, (t_L, k_L)\}$, where $L$ is the number of events in the sequence, $k_i \in \{1, 2..., M\}$ is the event type of the $i$th event, and $t_i$ is the time of $i$th event. For each event type $k$, a counting process $N_k(t)$ records the cumulative number of events that have occurred up until time $t$. The intensity function $\lambda_k(t)$ is defined as the expected instantaneous rate of type-$k$ events given the history of events, formalized as:

$$\lambda_k(t) = \frac{E[dN_k(t)|\mathcal{H}_t]}{dt}, \quad \mathcal{H}_t = \{(t_i, k_i)|t_i < t, k_i \in \{1...M\}\}$$

In a standard Hawkes process, which is a type of self-exciting multivariate point process, the intensity depends on the history of past events. The intensity function is defined as:

$$\lambda_k(t) = \mu_k + \sum_{k'=1}^{M} \int_0^t \phi_{k',k}(t-s)dN_{k'}(s) \tag{1}$$

$$= \mu_k + \sum_{(t_i, k_i) \in \mathcal{H}_t} \alpha_{k_i,k} \exp(-\delta_{k_i,k}(t-t_i)) \tag{2}$$

In this standard Hawkes processes, the impact functions $\phi_{k',k}(t-s)$ are assumed to follow an exponential decay. This intensity function provides insight into how likely it is for a specific event type to occur at any given moment in time, considering the past events up to time $t$.

Here, $\mu_k$ is the baseline intensity, and $\phi_{k',k}(t-s)$ is the impact function that quantifies how events of type $k'$ affect the intensity of events of type $k$. $\alpha_{k_i,k}$ controls the strength of the triggering effect, while $\delta_{k_i,k}$ determines how fast the influence decays over time.

## 3.2 IMPACT KERNEL SUB-NETWORK

In this work, we aim to relax the assumption that the impact functions follow a particular parametric form by introducing a neural network-based impact function. We begin by simplifying equation 1 for $t_j \in (0, T)$, where $T$ is the maximum observation time. The intensity function for the $k_i$-th event, $\lambda_{k_i}(t_j)$, is expressed as:

$$\lambda_{k_i}(t_j) = \mu_{k_i} + \sum_{i<j} \phi_{k_i,k_j}(t_j - t_i) \tag{3}$$

where $\mu_{k_i}$ is the base intensity for event type $k_i$, and $\phi_{k_i,k_j}(\Delta t)$ represents the *impact function* that quantifies the influence of an event of type $k_i$ occurring at time $t_j$ on the intensity $\lambda_{k_i}(t_j)$ at a later time $t_j$. This effect depends solely on the time difference $\Delta t = t_j - t_i$.

The total intensity $\lambda(t)$ for any event occurring at time $t$ is given by the sum of intensities over all event types, i.e., $\lambda(t) = \sum_{k=1}^{K} \lambda_k(t)$.

In our proposed approach, the impact functions $\phi_{i,j}(\Delta t)$ are modeled using an impact kernel $K(\Delta t)$ with $M^2$ outputs, where M is the number of event types. The kernel $K(\Delta t)$ is parameterized by a neural network that takes $\Delta t$ as input and outputs the impact relationships among all event type pairs. As shown in Figure 1, without event embedding, the impact function for each pair of event types is obtained by selecting the corresponding elements from the output of $K(\Delta t)$. The impact function can be written as:

$$\phi_{i,j}(\Delta t) = (e^{(i)})^\top K(\Delta t) e^{(j)} \tag{4}$$

Here, $K(\Delta t)$ encompasses all the impact kernels between event type pairs $(i, j)$ for $i, j \in \{1, 2...M\}$, and $e^{(i)}$ and $e^{(j)}$ are the corresponding one-hot vectors of the event types at times $t_i$ and $t_j$, respectively. The neural network used to parameterize $K(\Delta t)$ can have a simple or complex architecture. Although it is parameterized, we do not make any assumptions about the specific shape of the impact functions. Our experiments demonstrate that even simple neural networks are sufficient for modeling impact functions in real-world data. As the number of event types increases, the computational cost of modeling impact functions grows quadratically. However, if the $M * M$ kernel has a sparse structure, most event pairs have no impact to each other, and modeling all pairs becomes unnecessary. In such cases, embedding-based methods are more efficient, as discussed in the next section.

To optimize the model, we aim to maximize the log-likelihood $\ell(\mathcal{S})$ of an event sequence $\mathcal{S}$, which is given as follows (Mei & Eisner, 2017):

$$\ell(\mathcal{S}) = \sum_{i=1}^{L} \log \lambda_{k_i}(t_i) - \int_0^T \lambda(t)dt \tag{5}$$

The first term in equation 5 involves evaluating the impact function $\phi$ over $\mathcal{O}(L^2)$ time intervals. The second term, an integral, can be approximated using numerical methods or Monte Carlo integration. Although numerical methods may introduce bias depending on the approach, they tend to outperform Monte Carlo methods due to the latter's high variance. The numerical approximation can be represented as linear interpolation between observed events:

$$\int_0^T \lambda(t)dt \approx \sum_{j=2}^{L} \frac{(\lambda(t_j) + \lambda(t_{j-1}))(t_j - t_{j-1})}{2} \tag{6}$$

Alternatively, Monte Carlo integration estimates the integral as:

$$\int_0^T \lambda(t)dt = \sum_{j=2}^{L} \left( \frac{1}{N} \sum_{i=1}^{N} \lambda(t_{j-1,i}) \right) (t_j - t_{j-1}) \tag{7}$$

where $t_{j-1,i} \sim \text{Unif}(t_{j-1}, t_j)$ and $N$ is the number of samples drawn. Monte Carlo methods require $\mathcal{O}(L^2NK)$ evaluations of $\phi$, while numerical methods require $\mathcal{O}(LNK)$, making the latter more efficient for large datasets.

## 3.3 FORMULATING THE IMPACT KERNEL IN EMBEDDING SPACE

Let $W \in \mathbb{R}^{D \times M}$ represent the embedding matrix, where $M$ is the number of event types and $D$ is the embedding dimension. The embedding for event type $i$ is given by $We^{(i)}$, where $e^{(i)}$ is the standard basis vector selecting the $i$-th column of $W$. Similarly, the embedding for event type $j$ is $We^{(j)}$. The embedding matrix $W$ may be shared between the input and output events or may differ, i.e., $W_1 e^{(i)}$ for input and $W_2 e^{(j)}$ for output events.

For a given time interval between events, we define an impact kernel $K(\Delta t) \in \mathbb{R}^{D \times D}$ in the embedding space. In this formulation, the impact kernel is instantiated as a neural network with $D^2$ outputs. The impact of an event of type $i$ at time $t_i$ on the intensity of an event of type $j$ at time $t_j$ is expressed as:

$$\phi_{i,j}(t_j - t_i) = (We^{(i)})^\top K(t_j - t_i)(We^{(j)}) \tag{8}$$

Intuitively, for each event pair $(k_i, k_j)$, we select coloumns from the embedding matrix $W$ and compute a linear combination of the impact kernels, weighted by each event's contribution, to reconstruct the impact function for that pair. This method, referred to as the Embedded Neural Hawkes Process(ENHP), reduces the dimensionality of the impact kernels, making it well-suited for high-dimensional event spaces, while also capturing the sparse structure of event interactions. Additionally, the embedding matrix $W$ introduces interpretability by mapping events to latent topics, where each dimension corresponds to a distinct attribute. To ensure the non-negativity of the intensity, we apply a softplus function to all components, including $W$, $K(t)$, and $\mu_k$.

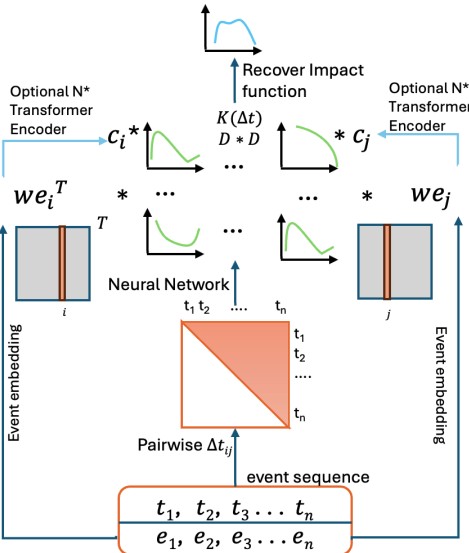

Figure 1: Architecture of the ENHP and ENHP-C. Without the event embedding and transformer encoder, the impact function follows formula4. Without the transformer encoder, it follows formula8. With all components, it corresponds to formula9.

We further enhance the model by introducing a transformer encoder on top of the event embeddings to capture contextualized event interactions. The contextualized embeddings generated by the transformer are given by:

$$\phi_{i,j}(t_j - t_i) = (c^{(i)})^\top K(t_j - t_i)(c^{(j)}) \tag{9}$$

where $c^{(i)}$ represents the output of the transformer encoder, which takes the sum of the event embedding and a temporal embedding as input. This extended model is termed the Embedded Neural Hawkes Process - Contextualized (ENHP-C). While this approach can potentially enhance performance by modeling more intricate event relationships and improving likelihood, the learned kernel impact functions become less interpretable, as the contextualized embeddings are hidden representations.

In this paper, $K(\Delta t)$ is parameterized by a neural network consisting of a fully connected layer with ReLU activation, followed by a linear output layer.

# 4 EXPERIMENTS

## 4.1 BASELINE

All the intensity-based generalized Hawkes process models mentioned in the related work section will be included as baseline methods, including NHP, THP, and SAHP. Additionally, we include the following two methods:

**Attentive Neural Hawkes Process** : AttNHP(Yang et al., 2022) introduces an attention mechanism into the traditional Neural Hawkes Process to enhance its ability to model event dependencies. In this approach, instead of relying solely on a recurrent neural network to capture the influence of past events, attention weights are applied to determine the relative importance of each historical event in predicting future occurrences. This makes the model more accurate and easier to understand because it focuses on the key events in the history.

**Fully Neural Network based Model for General Temporal Point Processes** FullyNN-TPP(Omi et al., 2019) uses a purely neural network-driven approach to model temporal point processes, replacing traditional hand-crafted intensity functions. This model leverages deep neural networks to parameterize the conditional intensity function of events, directly learning complex temporal dependencies. By employing flexible neural architectures, such as recurrent layers or temporal convolution, this method generalizes well across different types of event sequences without making strong assumptions about the underlying temporal dynamics.

## 4.2 REAL-WORLD DATASETS

To evaluate our model, we utilize several well-established datasets from different domains. Each dataset represents a sequence of events with time stamps and categorical labels defining the event types. With the exception of the MIMIC-IV dataset, results for baseline methods an all datasets are reproduced from previously published work (Mei & Eisner, 2017)(Xue et al., 2023).Below are the descriptions of the datasets used in our experiments:

**MIMIC-IV** The MIMIC-IV dataset contains comprehensive clinical data from patients admitted to the intensive care units (ICU) at a tertiary academic medical center in Boston. Specifically, we use procedure events, which include all medical procedures administered to patients during their ICU stay. Each procedure is time-stamped and categorized, representing different types of medical interventions. For this analysis, we included 84366 ICU stays with maximum sequence length of 240. There were $K = 159$ distinct event types (Johnson et al., 2023).

**Amazon** This dataset consists of time-stamped product review events collected from Amazon users between January 2008 and October 2018. Each event includes the timestamp of the review and the category of the product being reviewed, with each category mapped to a distinct event type. In this paper, we focus on a subset of the 5,200 most active users, where each user has an average sequence length of 70, and there are $K = 16$ event types (Ni et al., 2019).

**Retweet** This dataset captures sequences of user retweets, with events classified into three types based on the size of the user's following: "small" (under 120 followers), "medium" (under 1,363 followers), and "large" (more than 1,363 followers). A subset of 5,200 active users was extracted, with an average sequence length of 70 events per user and $K = 3$ event types (Zhou et al., 2013).

**Taxi** This dataset logs time-stamped taxi pick-up and drop-off events across New York City's five boroughs. Each event type is defined by the combination of the borough and whether the event is a pick-up or drop-off. This results in $K = 10$ event types. 2,000 drivers were randomly sampled with an average sequence length of 39 events (Whong, 2014).

**StackOverflow** This dataset tracks user activities on the StackOverflow platform, specifically the awarding of badges over two years. Each event corresponds to the awarding of a badge, with $K = 22$ different badge types. For this analysis, we use a subset of 2,200 active users, each with an average sequence length of 65 (Jure, 2014).

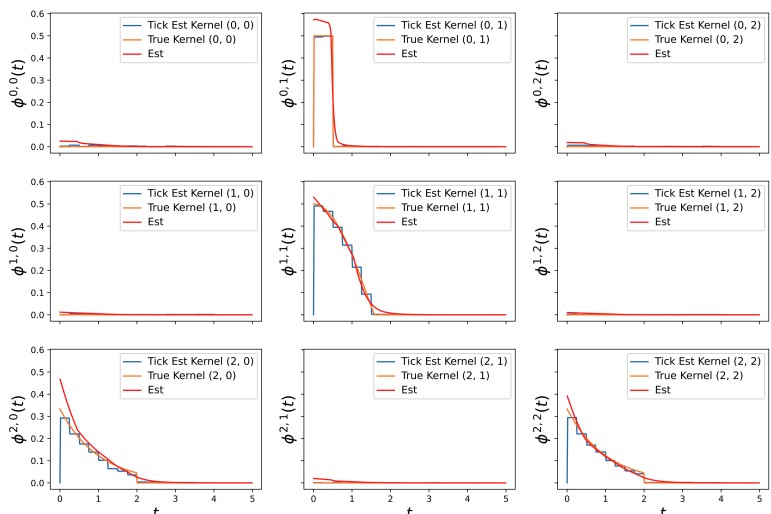

Figure 2: Fitted triggering kernel using ENHP

**MemeTrack** This dataset monitors the spread of "memes" (fixed phrases) across online news articles and blogs. It records time-stamped instances of meme usage from over 1.5 million websites, where each meme defines a distinct event type. The dataset includes $K = 5000$ event types and 80000 subjects were sampled. (Jure, 2014).

## 4.3 SIMULATION

We begin by verifying that our model successfully fits data generated from a known distribution, and that the impact kernels learned by our method match the known, true impact kernels. We generate a synthetic dataset using *tick*, an open-source machine learning library for Python that includes a Hawkes process module (Bacry et al., 2018). Specifically, a three-dimensional Hawkes process was generated with baseline intensities $\mu_0 = 0.3$, $\mu_1 = 0.05$ and $\mu_2 = 0.2$. The triggering kernels include four active (*i.e.*, nonzero) kernels: a step function, a cosine kernel, and two exponential kernels. Other kernels are inactive (*i.e.*, identically zero). Details are shown below:

$$\phi_{0,1} = \begin{cases} 0.5, & \text{if } t \in [0, 0.5] \\ 0, & \text{else} \end{cases} \quad \phi_{1,1} = cos(t/2) \quad \phi_{2,0} = exp(-x/3) \quad \phi_{2,2} = exp(-x/3)$$

The results are presented in Figure 2. Using the proposed ENHP, our method successfully recovers all four active impact functions, regardless of whether the function includes a jump point or is convex or concave. With the exception of the step function kernel, our approach outperforms the estimation from the tick library by providing more accurate shapes for the impact functions. In contrast, tick relies on step functions to approximate impact kernels, which can obscure the true form of the underlying functions. We did observe slight overestimation of the bump in the step function kernel, as well as minor overestimation in the inactive (constant-zero) kernel, but these could be improved by introducing regularization to the embedding matrix. Overall, our method proves effective in recovering various types of impact kernels in a multivariate Hawkes process.

## 4.4 PERFORMANCE ON BENCHMARK DATASETS

**Training detail** To compare the performance with other methods, we are using an open benchmark EasyTPP(Xue et al., 2023) to evaluate the other methods. EasyTPP is a newly developed benchmark framework for evaluating temporal point process models. It provides standardized datasets, evaluation metrics, and implementations of various baseline models, enabling fair and consistent comparisons across different methods. All hyperparameters for benchmark methods are provided by EasyTPP. For our method, all hyperparameters are included in the appendix. All likelihoods presented below are reported on the validation set. A single NVIDIA RTX graphics card was used to run all experiments.

**Likelihood comparison between purposed method** In this experiment, we compare the performance of the ENHP and ENHP-C in terms of log likelihood (LL). From table 1, we can find that

| Data | ENHP | ENHP-C |
|------|------|--------|
| Retweet | -3.67 ± 0.02 | -3.80 ± 0.02 |
| Amazon | -0.82 ± 0.03 | -0.82 ± 0.03 |
| Taxi | -0.24 ± 0.04 | -0.23 ± 0.05 |
| StackOverflow | -2.43 ± 0.07 | -2.36 ± 0.07 |
| MemeTrack | -10.97 ± 0.11 | -11.45 ± 0.12 |
| Simu | -1.80 ± 0.01 | -1.80 ± 0.01 |

Table 1: Performance comparison between ENHP and ENHP-C across datasets, with results presented as mean LL ± standard deviation.

the two methods perform similarly across most datasets, with the transformer-enhanced model occasionally underperforming compared to the basic embedding method, as seen in the Retweet and StackOverflow datasets. This suggests that for the real-world data we tested, our embedding-based approach is already sufficient to capture the necessary patterns without the need for the additional flexibility introduced by the inclusion of transformer layers. In addition, both models are also capable of handling datasets with a large number of event types, such as the Meme Track dataset, which contains 5,000 event types. Overall, while contextualizing embeddings using transformer layers may allow the model to capture more complex interactions, the additional benefit is not clearly reflected in the LL results on these real datasets. Given the large size of many of these datasets, we believe it is unlikely that this is the result of insufficient data, and more likely that ENHP is already sufficiently flexible to capture the underlying data distribution.

| Dataset | ENHP (Rank) | AttNHP (Rank) | NHP (Rank) | THP (Rank) | FullyNN (Rank) |
|---------|-------------|---------------|------------|------------|----------------|
| Retweet | **-3.665 (1)** | -4.164 (3) | -4.137 (2) | -4.560 (4) | -5.880 (5) |
| Taxi | -0.236 (3) | -0.227 (2) | **-0.208 (1)** | -0.442 (4) | -1.317 (5) |
| StackOverflow | -2.428 (2) | **-1.815 (1)** | -2.511 (4) | -2.468 (3) | -7.383 (5) |
| Amazon | -0.817 (2) | -2.560 (4) | **-0.720 (1)** | -2.388 (3) | -6.360 (5) |
| MIMIC-IV | -9.616 (2) | -9.904 (3) | **-8.634 (1)** | -15.942 (5) | -10.490 (4) |
| Average Rank | 2.0 | 2.6 | **1.8** | 3.8 | 4.8 |

Table 2: Performance comparison across datasets and models based on log-likelihood (LL). LL values are followed by the corresponding rank in parentheses. The last row shows the average rank across all datasets for each model.

**Likelihood Comparison between Our Method and Other Models** In our experiments, we primarily evaluated model performance using log-likelihood (LL). However, our main focus is on comparing our method, ENHP, with other mainstream models that incorporate parameterized assumptions. It is important to note that not all methods support the Meme track due to the high dimensionality of event types; therefore, we excluded it from this analysis. Additionally, we identified a configuration issue with SAHP in easyTPP, leading us to temporarily remove it from the comparison.

As shown in Table 2, our method demonstrates superior performance across multiple datasets. Specifically, on the Retweet dataset, ENHP achieves the best LL of -3.665, ranking first among all models. On the StackOverflow dataset, our method secures the second rank with an LL of -2.428, closely following AttNHP, which records an LL of -1.815. Similarly, in the Amazon and MIMIC-IV datasets, ENHP consistently ranks second with LL values of -0.817 and -9.616, respectively, narrowly trailing the top-performing model NHP.

On the Taxi dataset, although NHP achieves the lowest LL of -0.208, our method ranks third with an LL of -0.236, demonstrating competitive performance. Overall, ENHP attains an average rank of 2.0 across all datasets, which is very close to the best average rank of 1.8 achieved by NHP. These results indicate that our method's performance is consistently among the top performers across various datasets.

Similar patterns have been observed in previous studies (Xue et al., 2023; Yang et al., 2022), where transformer-based approaches (e.g., THP) do not consistently outperform other attention-based or neural network-based models. This aligns with our findings, where THP does not show significant performance improvements over other models. Additionally, while AttNHP outperforms NHP on the StackOverflow dataset—corroborating the conclusions presented in the AttNHP paper (Yang et al., 2022)—it performs worse than NHP on other datasets, consistent with results reported in the easyTPP paper for Taxi and Retweet. Our experimental results are in agreement with these findings, further validating the conclusions of our study.

Crucially, among all the methods compared, ENHP is the only model that offers complete interpretability. This means that while we achieve competitive performance in terms of LL—ranking first or second across most datasets—our model also provides strong interpretability. This combination of high performance and interpretability positions our method as a compelling choice for applications where understanding the model's decision-making process is essential.

### 4.5 INTERPRETATION OF RECOVERING IMPACT FUNCTION

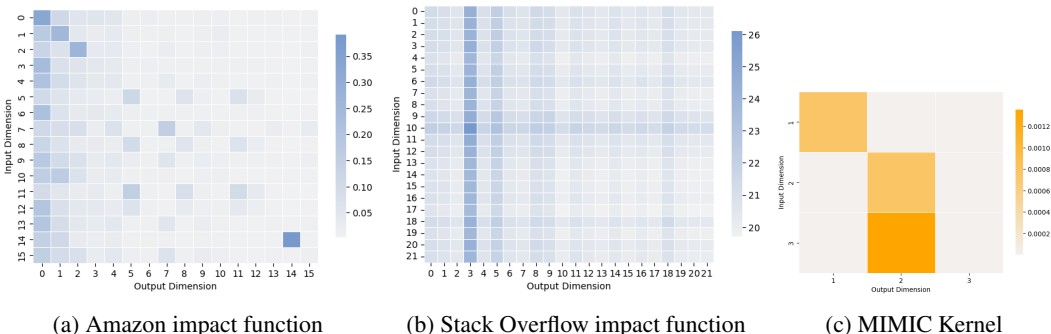

(a) Amazon impact function      (b) Stack Overflow impact function      (c) MIMIC Kernel

Figure 3: Heatmap of different dataset with regarding to impact function or kernel function

As demonstrated in the simulation, our method exhibits strong capability in recovering the impact functions between pairs of events. While it is possible to recover the entire impact function (i.e., intensity over time), this approach is not ideal for visualization due to the high dimensionality and sparse structure of most datasets. Therefore, we compute $\int \phi_{i,j}(t)dt$ for each impact function, which represents the cumulative impact from event $i$ to event $j$. This scalar value is then visualized through a heatmap.

**Amazon** The figure3a visualizes the learned impact function for the Amazon dataset, illustrating the influence between different product categories. Several key patterns emerge from this visualization: (1)Self-Excitation on the Diagonal: The darker squares along the diagonal indicate that several categories exhibit strong self-excitation. Notably, categories such as Type 14 (surf, skate, and street), Type 0 (clothing), Type 1 (shoes), and Type 2 (accessories) display high self-excitement. This means that customers are likely to purchase multiple items within the same category in a short period. For Type 14, it is common for customers buying surf or skate gear to need multiple specialized items, which explains the strongest self-excitation in this categories. Similarly, purchasing multiple pieces of clothing, shoes, or accessories in a single session aligns with typical consumer behavior patterns on Amazon, where shoppers often have extra saving for a regular subscription. (2) The first column, which corresponds to Type 0 (clothing), is notably darker across multiple rows. This indicates that purchases in other categories have a significant impact on clothing purchases. This makes sense given that clothing is a fundamental and prevalent category on this dataset and also in real Amazon sales.

**Stack Overflow** The figure 3b depicts the interactions between different events in the Stack Overflow dataset, where each event represents the awarding of a specific badge to a user. The results reveal that nearly all events tend to trigger Event 3 (Popular Question), followed by Event 5 (Nice Answer). The primary source of excitation is Event 10 (Notable Question), which has a strong influence on other events. A notable observation is that both Event 3 and Event 10 are the most frequent events in this dataset, which aligns with the behavior captured in the data. Our model successfully identifies these patterns, which not only match the known data facts but also reveal new insights into the influence structure within the event space. This suggests that our model is capable of uncovering meaningful and interpretable relationships between events.

### 4.6 INTERPRETATION OF RECOVERING KERNEL

Results on MIMIC-IV allow us to illustrate how our method facilitates interpretability. We begin by identifying specific procedures in the dataset that load most strongly on each embedding dimension (see Table 3). We see that the events that load most strongly on input embedding dimensions 1-3 are related to intubation and ventilation (Input 1) and obtaining culture results (Input 2-3), respectively.

In contrast, the events that load most strongly on output embedding dimensions 1-3 are related to X-Ray and EKG (Output 1), placement of intravenous and arterial lines (Output 2), and interventional radiology procedures, including catheterization and angiography (Output 3).

Referring to the corresponding impact kernel (see Figure 3c) then allows us to determine the impact of each input topic on each output topic. To effects are notable. First, we see that obtaining culture results (Input 2-3) is followed by an increase in the rate of line placement (Output 2), which may reflect preparation to administer intravenous antibiotics or other medications in response to positive culture. Second, we see that intubation and ventilation (Input 1) are are followed by an increase in rates of X-Ray and EKG (Output 1), which may reflect increased monitoring or verification of correct endotracheal tube placement.

Importantly, we have chosen a very small embedding dimension (3) to facilitate interpretation in this simplified example. However, this choice is not optimal to maximize performance, and prevents us from identifying more granular topics that may be more clinically impactful.

Table 3: Event Embedding for Topic Discovery

| Event Embedding for Input Events | | |
| --- | --- | --- |
| **Rank** | **Input 1** | **Input 2** | **Input 3** |
| **Topic Name** | **Ventilation** | **Cultures 1** | **Cultures 2** |
| 0 | Invasive Ventilation | Urine Culture | Nasal Swab |
| 1 | Intubation | Blood Cultured | Blood Cultured |
| 2 | Chest Tube Removed | Endoscopy | Urine Culture |
| 3 | Temporary Pacemaker Wires Discontinued | Family updated by RN | Foley Catheter |
| 4 | PA Catheter | 16 Gauge | Indwelling Port |
| 5 | PEG Insertion | Sputum Culture | Sputum Culture |
| 6 | Endoscopy | Indwelling Port | 16 Gauge |
| 7 | Percutaneous Tracheostomy | MAC | Extubation |
| 8 | Peritoneal Dialysis | Family updated by MD | Indwelling Port (PortaCath) |
| 9 | Extubation | Stool Culture | Stool Culture |
| **Event Embedding for Output Events** | | |
| **Rank** | **Output 1** | **Output 2** | **Output 3** |
| **Topic Name** | **X-Ray and EKG** | **Line Placement** | **Interventional Radiology** |
| 0 | Chest X-Ray | 20 Gauge | Interventional Radiology |
| 1 | EKG | 18 Gauge | Cardiac Cath |
| 2 | Blood Cultured | Invasive Ventilation | Temporary Pacemaker Wires Inserted |
| 3 | Family updated by RN | Arterial Line | Unplanned Extubation (non-patient initiated) |
| 4 | SEPS (Subdural Evacuating Port System) | Sevoflurane (Inhaled) | Angiography |
| 5 | Sevoflurane (Inhaled) | SEPS (Subdural Evacuating Port System) | Subdural Drain |
| 6 | Multi Lumen Cooling Catheter | VAC Change | Fall |
| 7 | Isoflurane (Inhaled) | ERCP (Done in unit) | Cath Lab (Sent) |
| 8 | Transferred to Floor | Pericardial Drain Removed | Surgical Procedure at Bedside |
| 9 | ICP - Camino | Blakemore / Minnesota Tube D/C | Small Bore Nasal Enteral Tube Placement |

## 5    CONCLUSION

In this work, we addressed the challenge of modeling event sequences with self-reinforcing dynamics by proposing a flexible Hawkes process model that maintains interpretability. Our approach leverages a neural impact kernel in event embedding space, allowing it to capture complex event dependencies without assuming specific parametric forms, while still retaining the core interpretability of traditional Hawkes processes.

We also introduced a mechanism to balance flexibility and interpretability by adding transformer encoder layers to contextualize event embeddings. However, our results showed that the flexible impact kernel alone is often sufficient to model the dynamics of real-world sequences, reducing the need for additional contextualization in most cases.

Overall, our method demonstrates competitive performance with existing models while maintaining interpretability, making it suitable for high-stakes applications where understanding event interactions is crucial. This balance between flexibility and interpretability can help inform decision-making and uncover temporal dependencies in various domains.

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

# A APPENDIX

## A.1 TRAINING HYPERPARAMETERS

Table 4: Hyperparameters used for the proposed method. "# head" and "# layer" are only applicable for contextualized embedding hyperparameters. "MC" stands for Monte Carlo integration, and "NC" stands for numerical integration.

| Dataset | # head | # layer | $D_{\text{model}}$ | Batch Size | Learning Rate | Solver |
|---|---|---|---|---|---|---|
| Retweet | 1 | 1 | 3 | 64 | $1 \times 10^{-3}$ | MC |
| Taxi | 1 | 1 | 10 | 256 | $1 \times 10^{-4}$ | MC |
| StackOverflow | 2 | 2 | 22 | 64 | $1 \times 10^{-3}$ | MC |
| Amazon | 2 | 2 | 16 | 64 | $1 \times 10^{-3}$ | MC |
| MIMIC-IV | 2 | 2 | 25 | 32 | $1 \times 10^{-2}$ | NC |
| MemeTrack | 2 | 2 | 50 | 256 | $1 \times 10^{-2}$ | NC |
| Simulation | 1 | 1 | 3 | 128 | $1 \times 10^{-4}$ | MC |

## A.2 EXPERIMENT: DIFFERENT MODEL DIMENSION VS LIKELIHOOD

In this experiment, we evaluate the performance of models on the MIMIC-IV and Meme datasets by varying the embedding dimensions. The original event dimension of MIMIC-IV is 159, while that of Meme is significantly larger at 5,000. As expected, the reduction in log-likelihood for MIMIC-IV remains relatively small when the dimension is reduced below 10, whereas Meme experiences a more pronounced decline. This is reasonable, given that Meme's higher original dimension suggests more potential active relationships between event types. Despite this, both datasets achieve strong log-likelihood performance in lower dimensions (10-50), indicating that many event pairs in the original impact kernel dimension are likely unrelated. This further suggests that the original impact kernel may have a sparse structure. Our method effectively recovers the impact function even in low dimensions, highlighting its ability to capture the essential relationships while reducing dimensional complexity.

However, as shown in the figure4, there is a noticeable drop in performance for MIMIC-IV when the dimension is set to 40. A potential reason for this decline could be the relatively long maximum sequence length in the MIMIC-IV dataset, which increases the GPU memory requirement for computation. As a result, the batch size had to be limited to 16, which might cause instability in training. Since the sequences in MIMIC-IV vary greatly in length, with some batches containing very short sequences, this variability could lead to an unstable training process, ultimately affecting the model's performance.

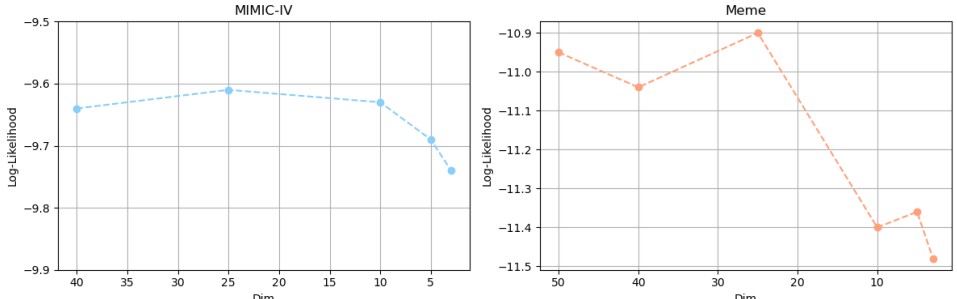

Figure 4: Different dimension vs Likelihood

