# OpenReview forum: "Hawkes process revisited: balancing interpretability and flexibility with contextualized event embeddings and a neural impact kernel"
_ICLR.cc/2025/Conference — ICLR 2025 Conference Withdrawn Submission_

### Official Review · Reviewer_NN4h · 2024-10-31

**Soundness:** 3
**Presentation:** 3
**Contribution:** 1
**Rating:** 3
**Confidence:** 4

**Summary:**

This paper reformulates the classic Hawkes process by incorporating neural networks to enhance its expressive power. Specifically, the authors propose three types of neural Hawkes processes: one based on one-hot vectors, one based on event representations, and one utilizing latent vector representations. Through extensive experiments, the authors demonstrate that the event representation-based neural Hawkes process generally achieves strong predictive performance while maintaining excellent interpretability.

**Strengths:**

- The paper provides a detailed analysis of the interpretability of the proposed model, as discussed in Sections 4.5 and 4.6.
- The writing is clear, making the paper easy to follow, and the results straightforward to reproduce.

**Weaknesses:**

The paper lacks significant innovation. The authors should refer to Equation 11 in reference [1] and Equations 3, 4, and 5 in reference [2]. The approach in this paper closely mirrors these works, specifically the use of neural networks to parameterize the impact kernel of the Hawkes process.

1] Song Y, Lee D, Meng R, et al. Decoupled Marked Temporal Point Process using Neural Ordinary Differential Equations. In The Twelfth International Conference on Learning Representations.
[2] Zhou Z, Yu R. Automatic Integration for Fast and Interpretable Neural Point Processes. In Learning for Dynamics and Control Conference. PMLR, 2023: 573-585.

**Questions:**

- In line 215, the authors state that the computational complexity of the Monte Carlo method is $O(L^2NK)$, while the computational complexity of the numerical method is $O(LNK)$. Is there an error here? Based on my understanding, the computational complexity of the numerical method should be $O(L^2K)$.

- In line 133, the authors claim that SAHP does not explicitly model decaying temporal effects. However, I would argue that their model also does not capture decaying temporal effects. Specifically, in line 271, the authors cannot ensure that the output $K$ decreases as $\Delta t$ increases.

- How were the results in Figure 3c obtained? From my understanding, the "dimension" in Figure 3c represents "topics" (which includes multiple event types, as described in Table 3), whereas the "dimension" in Figures 3a and 3b pertains to "event types."

---

### Official Review · Reviewer_bWrm · 2024-11-01

**Soundness:** 2
**Presentation:** 3
**Contribution:** 2
**Rating:** 5
**Confidence:** 4

**Summary:**

This paper focuses on understanding and improving the trade-off between the flexibility and interpretability of the Hawkes process. The authors replace the Hawkes process's parametric kernel functions with a neural network-based impact kernel within an event embedding space, thereby enhancing the model’s flexibility. This neural network-based impact kernel retains some properties of the Hawkes process, such as positive intensity and additive influence, thereby enabling good interpretability. Additionally, to manage the balance between model complexity and interpretability, the authors introduce optional transformer encoder layers to contextualize event embeddings.

**Strengths:**

1. The motivation is clear, and the issue of enhancing the interpretability of the neural Hawkes process is of considerable significance.

2. This paper proposes three designs for neural kernel functions, each balancing model flexibility and interpretability to different extents.

3. The article is well-structured and easy to follow

**Weaknesses:**

1. There are generally three metrics for evaluating point process models: log-likelihood, accuracy (acc), and root mean square error (RMSE) [1]. Among these, log-likelihood measures the model’s goodness-of-fit, while accuracy and RMSE measure the model’s event prediction performance. This paper only uses log-likelihood. Furthermore, in terms of log-likelihood, the proposed method does not demonstrate a significant advantage over other baseline models.

2. Equation 8 seems to imply an assumption that the influence between events is always a positive excitation (because the softplus function is applied to all components, including W , K(t), and μ_k). What if the influence of events is "inhibition" rather than "excitation"?

3. The neural kernel function seems capable of modeling only the influence of one event on another, but in some scenarios, multiple events occurring together may be required to trigger a subsequent event, as in the case of synergy [2].

Reference:

[1] EASYTPP: TOWARDS OPEN BENCHMARKING TEMPORAL POINT PROCESSES (ICLR'24)

[2] CAUSE: Learning Granger Causality from Event Sequences using Attribution Methods (ICML'20)

**Questions:**

Please refer to Strengths and Weaknesses for more details.

---

### Official Review · Reviewer_LgmM · 2024-11-02

**Soundness:** 2
**Presentation:** 2
**Contribution:** 2
**Rating:** 3
**Confidence:** 5

**Summary:**

The paper proposes a novel approach for modeling Hawkes processes (HPs), where a deep neural network is used to model the influence function, making it not only more flexible than traditional HPs but also enhancing model interpretability.

**Strengths:**

1. The paper provides a comprehensive overview and classification of current models for event sequence prediction, covering traditional HPs, RNN-based HPs, attention-based HPs, and so on.
2.  This paper is easy to follow.

**Weaknesses:**

1. This paper is an incremental work of TPPs. There are many related works to investigate the interpretability of TPPs.  Adopting neural networks to model impact kernels are also quite common. The core contribution of model novelty is limited.
2. The assumption that the influence between events is always positive is too strong, and many real-world scenarios  do not fit this assumption, which limits the model’s flexibility.
3. The experimental improvement is not evident. Compared with current methods, the improvement of this work is marginal.

**Questions:**

See above weaknesses.

---

### Official Review · Reviewer_rZ9z · 2024-11-04

**Soundness:** 2
**Presentation:** 2
**Contribution:** 1
**Rating:** 3
**Confidence:** 4

**Summary:**

Neural network-based HPs offer greater flexibility and improved performance in modeling event sequences with self-reinforcing dynamics, but at the cost of interpretability.  This paper proposes to address this challenge by leveraging a neural impact kernel in event embedding space, which allows to capture complex event dependencies without assuming specific parametric forms, while still retaining the core interpretability of traditional Hawkes processes. Real data experiments are conducted to demonstrate the competitive performance with existing models while maintaining interpretability.

**Strengths:**

(1) Introduce a generalized Hawkes process where impact functions are defined via a flexible, neural network-based impact kernel within an event embedding space.

(2) The proposed method is flexible to incorporate transformer encoder layers to contextualize event embeddings based on the historical sequence of events, which can explicitly manage the balance between interpretability and model complexity.

(3) The authors show that the transformer encoder layers are often unnecessary to achieve state-of-the-art performance and demonstrates the competitive performance of proposed method with existing models while maintaining interpretability with real data experiments

**Weaknesses:**

(1) The core idea is simple, which introduces a neural network-based impact kernel within an event embedding space to improve interpretability while keeping competitive performance. It would be better to diccuss the effects of the impact kernel on the modeling performance in details, and also illustrate how to choose or design the appropriate kernels in applications for better balance between interpretability and model complexity.

(2) Generally, increased model complexity may lead to higher model likelihood value. So, it's not adequate to only compare the likelihood between the proposed method and other models. It would be necessary to also compare the out-of-sample metrics such as out-of-sample prediction performance and so on, for all the related comparisons between the proposed method including the variant with transformer encoder layer and existing methods.

(3) The authors states that "Given the large size of many of these datasets, we believe it is unlikely that this is the result of insufficient data, and more likely that ENHP is already sufficiently flexible to capture the underlying data distribution" in line 395-397 of page 8. The statement is not adequate and convincing, and it's better to add some necessary experiments based on simulated data for illustration.

**Questions:**

See the Weaknesses.

**Details Of Ethics Concerns:**

-

---

### Note · Authors · 2024-11-18

I have read and agree with the venue's withdrawal policy on behalf of myself and my co-authors.